# The complement C3-complement factor D-C3a receptor signalling axis regulates cardiac remodelling in right ventricular failure

Shogo Ito[1,2], Hisayuki Hashimoto [1,3], Hiroyuki Yamakawa[1,3], Dai Kusumoto[1,3], Yohei Akiba [1], Takahiro Nakamura[1], Mizuki Momoi[1], Jin Komuro[1], Toshiomi Katsuki[1], Mai Kimura[1], Yoshikazu Kishino[1], Shin Kashimura[1], Akira Kunitomi[1], Mark Lachmann[1], Masaya Shimojima[1], Gakuto Yozu[1], Chikaaki Motoda[1], Tomohisa Seki[1,4], Tsunehisa Yamamoto[1], Yoshiki Shinya[1], Takahiro Hiraide[1], Masaharu Kataoka[1,5], Takashi Kawakami[1], Kunimichi Suzuki [6], Kei Ito[7], Hirotaka Yada [7,8], Manabu Abe [9], Mizuko Osaka[10], Hiromi Tsuru[10], Masayuki Yoshida[10], Kenji Sakimura[9], Yoshihiro Fukumoto [2], Michisuke Yuzaki[6], Keiichi Fukuda [1] & Shinsuke Yuasa [1]✉

Failure of the right ventricle plays a critical role in any type of heart failure. However, the mechanism remains unclear, and there is no specific therapy. Here, we show that the right ventricle predominantly expresses alternative complement pathway-related genes, including *Cfd* and *C3aR1*. Complement 3 (*C3*)-knockout attenuates right ventricular dysfunction and fibrosis in a mouse model of right ventricular failure. C3a is produced from C3 by the C3 convertase complex, which includes the essential component complement factor D (Cfd). *Cfd*-knockout mice also show attenuation of right ventricular failure. Moreover, the plasma concentration of CFD correlates with the severity of right ventricular failure in patients with chronic right ventricular failure. A C3a receptor (C3aR) antagonist dramatically improves right ventricular dysfunction in mice. In summary, we demonstrate the crucial role of the C3-Cfd-C3aR axis in right ventricular failure and highlight potential therapeutic targets for right ventricular failure.

Cardiovascular disease is the leading cause of death worldwide[1]. Substantial efforts have been made in studying the pathophysiology of heart failure due to its high prevalence and poor prognosis. Most of the previous studies have focused on left ventricular (LV) failure since the left ventricle plays a central role in supplying blood to the systemic circulation against strong resistance[2]. Therefore, several medical therapies for heart failure have been developed, such as beta-blockers, angiotensin-converting enzyme inhibitors, and mineralocorticoid

receptor antagonists. Right ventricular (RV) failure has not been a central issue because the right ventricle pumps blood solely to the pulmonary artery against weak resistance[3]. However, accumulating evidence indicates that RV failure significantly contributes to the pathophysiology of various types of heart failure[4,5]. RV failure is one of the most important predictors of symptoms and prognosis in patients with pure right-sided heart disease[6]. Additionally, RV failure is a strong independent risk factor for survival in patients with LV failure[7,8].

However, drugs developed for treating LV failure are not effective against RV failure[9], and no effective therapies specifically targeting RV failure exist[9].

Hence, we aimed to understand the RV failure pathophysiology and develop an effective treatment for RV failure. The right ventricle differs from the left ventricle in many ways, including its structure, function, and developmental origin[10–12]. In terms of cellular properties, cardiomyocytes in the right ventricle are different from those in the left ventricle[10,11]. In humans, certain diseases predominantly affect the right ventricle[13,14]. These findings indicate that the right ventricle has unique characteristics, and hence, there exist therapeutic targets specific to RV failure. Here, we focussed on the right ventricle to dissect the unique molecular signature of RV failure and develop a specific therapy. We believe our findings could pave way for development of RV failure-specific therapeutics.

## Results

### Complement system activation in the right ventricle
To understand the molecular signature of the right ventricle, we first screened for genes specifically expressed in the right ventricle. We separated the murine heart tissue into the left ventricle, right ventricle, and ventricular septum and performed transcriptome analysis. Each cardiac tissue showed differential gene expression patterns for cardiac-specific genes (Supplementary Fig. 1a–f). Global gene expression analysis indicated that each cardiac region showed similar but different gene expression patterns (Fig. 1a). Ingenuity pathway analysis (IPA) of the differentially expressed genes between the right ventricle and left ventricle or between the right ventricle and ventricular septum showed that genes involved in the complement system were significantly enriched in the right ventricle, represented by *Cfd*, *C3*, and *C3ar1* (Fig. 1b, c). Importantly, *CFD*, *C3*, and *C3AR1* are also highly expressed in human RV (Fig. 1d–f). During complement activation, the C3 protein is cleaved into C3a and C3b by C3 convertase complex factors, including complement factor D (Cfd), and C3aR is the receptor of C3a. Therefore, we surmised that the complement system might play an important role in RV functions. To investigate whether the complement system was altered under pathological conditions, we generated an RV failure mouse model induced by pulmonary artery constriction (PAC)[15]. Interestingly, PAC also increased the expression of *Cfd* and *C3ar1* in the right ventricle (Fig. 1g–i). We then confirmed RV dysfunction and upregulation of cardiac failure and fibrotic markers specifically in the right ventricle (Fig. 1j, k; Supplementary Fig. 1g–j). Since C3a is hardly detectable in histological analysis and is a cleaved fragment of C3, C3d is used as a marker of C3 activation and is covalently fixed to tissues[16]. C3d was detected in the right ventricle only after PAC (Fig. 1l). These data suggest that the complement system was activated in the right ventricle, and it might contribute to the pathophysiology of RV failure.

### C3 is a critical factor for RV failure development
C3 is a central component of the complement system, and its activation stimulates several downstream pathways[17]. To explore the possibility that depletion of C3 modulated the progression of RV failure, we performed PAC in wild type (WT) and C3$^{-/-}$ mice. Interestingly, RV systolic dysfunction and dilatation induced by PAC were attenuated in C3$^{-/-}$ mice compared with WT mice, but left ventricle was not affected (Fig. 1j, k; Supplementary Fig. 1k–m). Hemodynamic study using a micro-catheter placed in both ventricles also showed RV dysfunction in WT mice after PAC, represented by a greater RV end-diastolic pressure, but not in C3$^{-/-}$ mice after PAC (Fig. 1m; Supplementary Fig. 1n). Additionally, after PAC, C3$^{-/-}$ mice showed a reduction in the fibrotic area, cardiac failure, and fibrotic marker gene expression in the right ventricle compared with WT mice (Fig. 1p–s; Supplementary Fig. 1o, p).

To investigate whether C3 was also involved in the development of LV failure, we performed transverse aortic constriction (TAC) in WT and C3$^{-/-}$ mice. TAC induced LV systolic dysfunction, dilatation, and fibrosis in the WT and C3$^{-/-}$ mice (Supplementary Fig. 2a–d). Interestingly, TAC also mildly induced fibrosis and elevation of cardiac failure and fibrotic markers in the right ventricle of WT mice. These pathological changes were attenuated in the right ventricle of C3$^{-/-}$ mice (Supplementary Fig. 2c–i). These data suggest that C3 contributed to the development of heart failure specifically in the right ventricle.

### The liver-derived C3 plays a critical role in RV failure development
Although plasma C3 is mainly produced in the liver, C3 is also produced in other tissues where it functions locally[18,19]. To clarify the origin of C3, which contributed to the development of RV failure, we generated a C3 floxed mouse model by conventional homologous recombination (Supplementary Fig. 3a, b). We then created the cardiac- or liver-specific *C3* knockout mice by crossing C3 floxed mice with α-myosin heavy chain promoter-driven Cre (αMHC-Cre) or albumin promoter-driven Cre (Alb-Cre) mice, respectively, and performed PAC surgery. Intriguingly, PAC induced RV dysfunction and dilatation in the cardiac-specific *C3* knockout mice, but these changes were attenuated in the liver-specific *C3* knockout mice (Fig. 2a, b; Supplementary Fig. 3c–e). RV fibrosis was attenuated only in the liver-specific *C3* knockout mice, but left ventricle was not affected (Fig. 2c–f; Supplementary Fig. 3f–i). Consistently, cardiac failure and fibrotic markers were significantly upregulated in the cardiac-specific *C3* knockout mice compared with the liver-specific *C3* knockout mice (Fig. 2g–j). These data suggest that the liver-derived C3 plays a crucial role in the development of RV failure.

### Cfd/CFD is a critical factor for RV failure
In the complement activation cascade, cleavage of C3 by C3 convertase generates the active molecules C3a (anaphylatoxin) and C3b (opsin)[20]. Cfd is required for the generation of the C3 convertase complex and is essential for producing C3a[21]. Since *Cfd* was enriched in the right ventricle, we next focused on the C3-Cfd signalling axis in the pathogenesis of RV failure. We performed PAC in *Cfd*$^{-/-}$ mice, which showed no apparent gross abnormalities at baseline[22,23]. Similar to C3$^{-/-}$ mice, PAC-induced RV dysfunction and dilatation were significantly suppressed in *Cfd*$^{-/-}$ mice, but left ventricle was not affected (Fig. 3a, b; Supplementary Fig. 4a–c). RV fibrosis, cardiac failure, and fibrotic marker gene expression were also significantly suppressed in *Cfd*$^{-/-}$ mice compared with WT mice (Fig. 3c–g; Supplementary Fig. 4d, e).

Based on the findings that the C3-Cfd signalling axis contributed to the development of RV failure in mice, we decided to examine whether the complement system was also involved in human RV failure pathogenesis. We examined the plasma concentrations of C3 and CFD in patients with chronic RV failure and compared these with the severity of the disease represented by the mean pulmonary artery (PA) pressure and plasma B-type natriuretic peptide (BNP) level. Plasma C3 concentration did not correlate with plasma BNP levels or mean PA pressure (Supplementary Fig. 4f, g). However, plasma CFD concentration was significantly correlated with plasma BNP levels and mean PA pressure (Fig. 3h, i). These data suggest that CFD was correlated with the severity of human RV failure.

### C3a directly regulates cardiac gene expression
Since *C3ar1* was highly expressed in the right ventricle, we focused on the C3a-C3aR signalling axis in the heart. We examined whether C3a regulated cardiac-specific gene expression in cultured rat cardiomyocytes. Several types of mitogen-activated protein kinases (MAPKs) play a central role in cardiac remodelling and heart failure[24]. The addition of

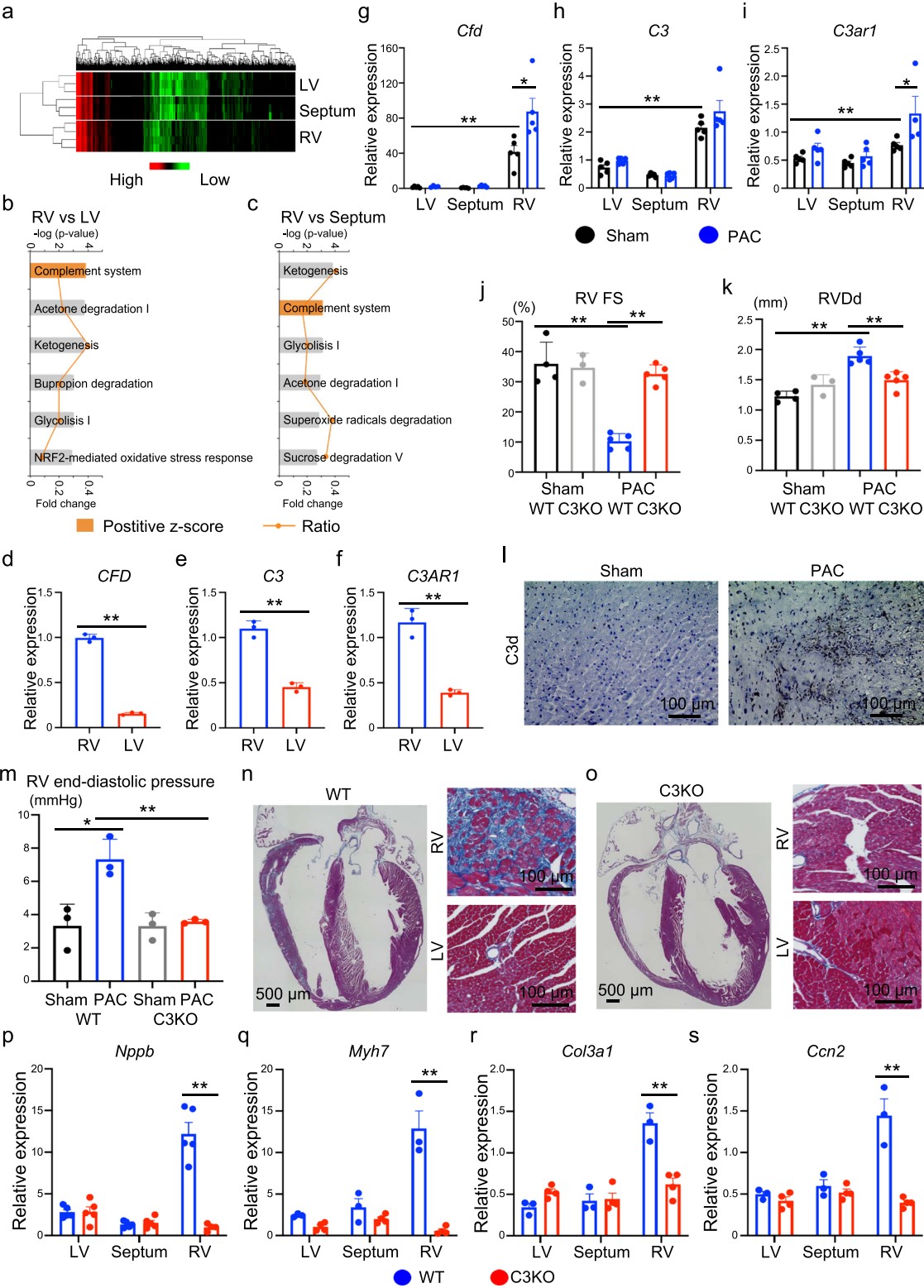

C3a activated extracellular signal-regulated kinase (ERK) in a time-dependent manner and weakly activated p38, but it did not activate c-Jun N-terminal kinase (JNK) (Fig. 4a). To examine whether C3a-dependent ERK phosphorylation was mediated by C3aR, we used siRNA to knockdown *C3ar1*. As expected, *C3ar1* knockdown blocked C3a-dependent ERK activation (Fig. 4b). Next, to understand the role of C3a in cardiomyocytes, we performed a global gene expression analysis of cardiomyocytes cultured in the presence or absence of C3a stimulation (Fig. 4c; Supplementary Fig. 5a). IPA of genes enriched in

the presence of C3a stimulation showed that the heart failure- and inflammation-related pathways were involved (Fig. 4d). We then searched for genes that showed changes in expression similar to that observed in our in vivo analyses. Among the genes enriched under C3a stimulation (Supplementary Fig. 5a), the expression of *Ccl5* and *Cebpa* was significantly upregulated in WT mice with PAC and suppressed in C3$^{-/-}$ mice with PAC (Supplementary Fig. 5b, c). Therefore, these genes could be the downstream targets of the C3a-C3aR signalling axis in RV failure.

**Fig. 1 | Global depletion of complement factor C3 attenuates right ventricular remodelling in pulmonary artery constriction (PAC) mouse model. a** Global gene expression heatmap for differentially expressed genes in the left ventricle (LV), ventricular septum, and right ventricle (RV) ($n = 3$–4). **b, c** Ingenuity pathway analysis of differentially expressed genes. Significance of the association between the dataset and the canonical pathway (−log ($p$-value) and fold change), and the $z$-score prediction are shown. The significance values are calculated by the right-tailed Fisher's exact test. **d–f** mRNA expression of *CFD*, *C3*, and *C3AR1* in human RV and LV ($n = 3$, $p = 0.0004$, $p < 0.0001$, $p = 0.001$). Data are presented as mean ± standard error of the mean (SEM). **g–i** mRNA expression of *Cfd*, *C3*, and *C3ar1* in sham-operated ($n = 4$–5, $p = 0.0006$, $p < 0.0001$, $p = 0.003$) and PAC model mice ($n = 4$–5, $p = 0.024$, $p = 0.1798$, $p = 0.0207$). Data are presented as mean ± SEM. **j, k** Measured values of the echocardiogram in sham and PAC models of wild type (WT) and *C3* knockout (C3KO) mice ($n = 3$–5, **j** $p = 0.001$, $p < 0.0001$. **k** $p < 0.0001$, $p = 0.002$). The RV contractile function (right ventricular fractional shortening [RV FS]) and RV size (right ventricular end-diastolic diameter [RVDd]) were evaluated. Data are presented as mean ± standard deviation (SD). **l** Representative immunostaining images for C3d in the RV of PAC model and sham-operated mice ($n = 3$). **m** Measured values of the catheter analysis in sham and PAC models of WT and C3KO mice ($n = 3$, $p = 0.0174$, $p = 0.0059$). RV end-diastolic pressure was evaluated. **n, o** Representative Azan staining images of the heart in WT and C3KO mice with PAC ($n = 3$). **p–s** mRNA expression of *Nppb*, *Myh7*, *Col3a1*, and *Ccn2* in the LV, ventricular septum, and RV of WT and C3KO mice with PAC ($n = 3$–4, $p = 0.00194$, $p = 0.0010$, $p = 0.0029$, $p = 0.0017$). Data are presented as mean ± SEM. mRNA expression of target genes was normalised to that of *Gapdh*. Significance was assessed using a two-tailed unpaired Student's $t$-test. *$p < 0.05$; **$p < 0.01$.

## C3aR blockade ameliorates the development of RV failure after PAC

From a clinical point of view, we investigated the applicability of the C3a-C3aR pathway in treating RV failure. We used a non-peptide C3aR antagonist, SB290157, to selectively block C3aR signalling[25]. SB290157 administration significantly suppressed the development of RV dysfunction after PAC in WT mice (Fig. 4e, f; Supplementary Fig. 6a–c). Furthermore, increases in the weights of the liver, lung, and heart, which are signs of heart failure, were suppressed by SB290157 treatment (Supplementary Fig 6d–f). Fibrosis and elevation of cardiac failure and fibrotic markers in the right ventricle were also attenuated in mice with PAC by SB290157 treatment, but left ventricle was not affected (Fig. 4g–j; Supplementary Fig. 6g–k).

Given that arrhythmia could be a fatal complication of RV failure, we investigated whether SB290157 affected the incidence of arrhythmia in mice with PAC. RV free wall tachypacing easily induced ventricular tachyarrhythmia in mice with PAC, but not in sham-operated mice. Interestingly, $C3^{−/−}$ and WT mice with SB290157 treatment significantly suppressed the induction of ventricular tachyarrhythmia and reduced the duration of ventricular tachyarrhythmia in mice with PAC (Fig. 4k–m). Calcium homeostasis plays a pivotal role in cardiomyocyte excitation-contraction coupling, and its impairment leads to RV failure and arrhythmic events[26]. In isolated RV cardiomyocytes, time to peak $Ca^{2+}$ transient was prolonged by PAC in WT mice, and its prolongation was suppressed in C3KO mice (Fig. 4n–o). The frequency of spontaneous $Ca^{2+}$ waves was increased by PAC in WT RV cardiomyocytes, and its frequency was attenuated in C3KO RV cardiomyocytes (Fig. 4p–q). Therefore, these data suggest that abnormal $Ca^{2+}$ dynamics is manifested in RV failure, and C3KO ameliorates its aberrant $Ca^{2+}$ dynamics.

## Discussions

RV functioning integrates preload, afterload, contractility, configuration, size, pericardial constraint, and interaction with the left ventricle[27]. Dysregulation of each of these factors induces RV failure. Its underlying causes include pulmonary vascular disease, parenchymal lung disease, RV infarction, congenital heart diseases, and LV failure. Although therapies for some specific diseases have been developed, specific therapies for RV failure have not been developed. To develop a drug for RV failure, it is important to understand its pathogenesis by murine RV failure models[28,29]. Constriction of the pulmonary artery could induce less systemic or toxic effects, but a constant constriction of the pulmonary artery would ensure a constant afterload[30,31]. Although PAC operation is technically difficult in mice and requires time to master, a well-sized PAC represents a valuable model of chronic pressure overload-induced RV failure.

Understanding innate immunity helps us to understand the role of the complement system in sterile inflammation[32]. Recent data show the possible involvement of the complement system in LV failure[33]. The complement system is activated in patients with LV failure, and complement factors are associated with clinical outcomes[34,35]. Low levels of plasma C3 can be a predictive marker of early death in patients with LV failure[36], and increased levels of complement factor B and Bb are associated with mortality in patients with LV failure[37]. Animal model studies also uncovered the role of the complement system in LV failure[38–40]. Pulmonary arterial hypertension is one of the important diseases to induce RV failure and is induced by hypoxia in experimental animals. The complement system activation was observed in the perivascular lesion in hypoxic lung, and would be involved in the pathology of pulmonary hypertension[41,42]. These basic and clinical studies focused solely on LV failure and lung, and not on RV failure.

Upon activation of the complement system, the central component factor C3 is cleaved[43]. Cfd is a rate-limiting enzyme in the alternative pathway and controls subsequent processes[44]. In the right ventricle, *Cfd* is predominantly expressed. C3 is highly abundant, but C3a is quickly converted by carboxypeptidase N into C3a-desArg[45] and C3a-desArg cannot bind to C3aR[46]. Therefore, we speculated that the C3a-C3aR signalling axis was only involved in RV failure. C3a-mediated signalling has been described as proinflammatory or anti-inflammatory; thus, these effects are context-specific[47–49]. It remains unclear why complement-related genes are highly expressed in the right ventricle. The complement system fundamentally prepares the body for infections by a lytic cascade activation, but is a complex innate immune surveillance system, playing a role in host homeostasis, inflammation, and in the defense against pathogens[50,51]. The right ventricle is in the venous return circulation system and is possibly at a risk of pathogen and foreign material intrusion. Therefore, basal expression of complement-related genes may play a role in surveillance and protection under physiological conditions. Nonetheless, in the current study, it remains unclear whether the upstream complement pathway could be activated and as to how this pathway is maintained in RV failure. The C3aR antagonist rescued the pathological phenotype, and we speculated that the downstream pathway, the membrane attack complex, would not play a role in RV failure. Therefore, it would be interesting for future studies to determine whether the downstream signalling could be activated in RV failure. The C3aR antagonist, SB290157 is widely used to explore the role of C3aR in animal experiments, but may have other effects, such as a partial C5aR1 agonist[52]. In drug study, we should take off target effect into consideration.

In summary, we showed that the right ventricle specifically expressed *Cfd* and *C3aR1*, and their expression was exaggerated in PAC-induced RV failure. Whole body *C3* deletion, *Cfd* deletion, and the liver-specific *C3* deletion ameliorated PAC-induced RV failure in mice. In patients with RV failure, the CFD concentration was significantly correlated with the severity of RV failure. C3a directly regulated the expression of several genes through C3aR in cardiomyocytes, and the C3aR antagonist ameliorated PAC-induced RV failure in mice. Currently, there is no specific biomarker for patients with RV failure, and hence, we speculate that CFD could be a disease marker for RV failure in humans. Anaphylatoxins have recently been explored as therapeutic targets for several diseases[53–55]. Here, we showed the crucial role of the C3-Cfd-C3aR signalling axis in RV failure and highlighted the potential therapeutic targets for RV failure, which has no pharmacologic options

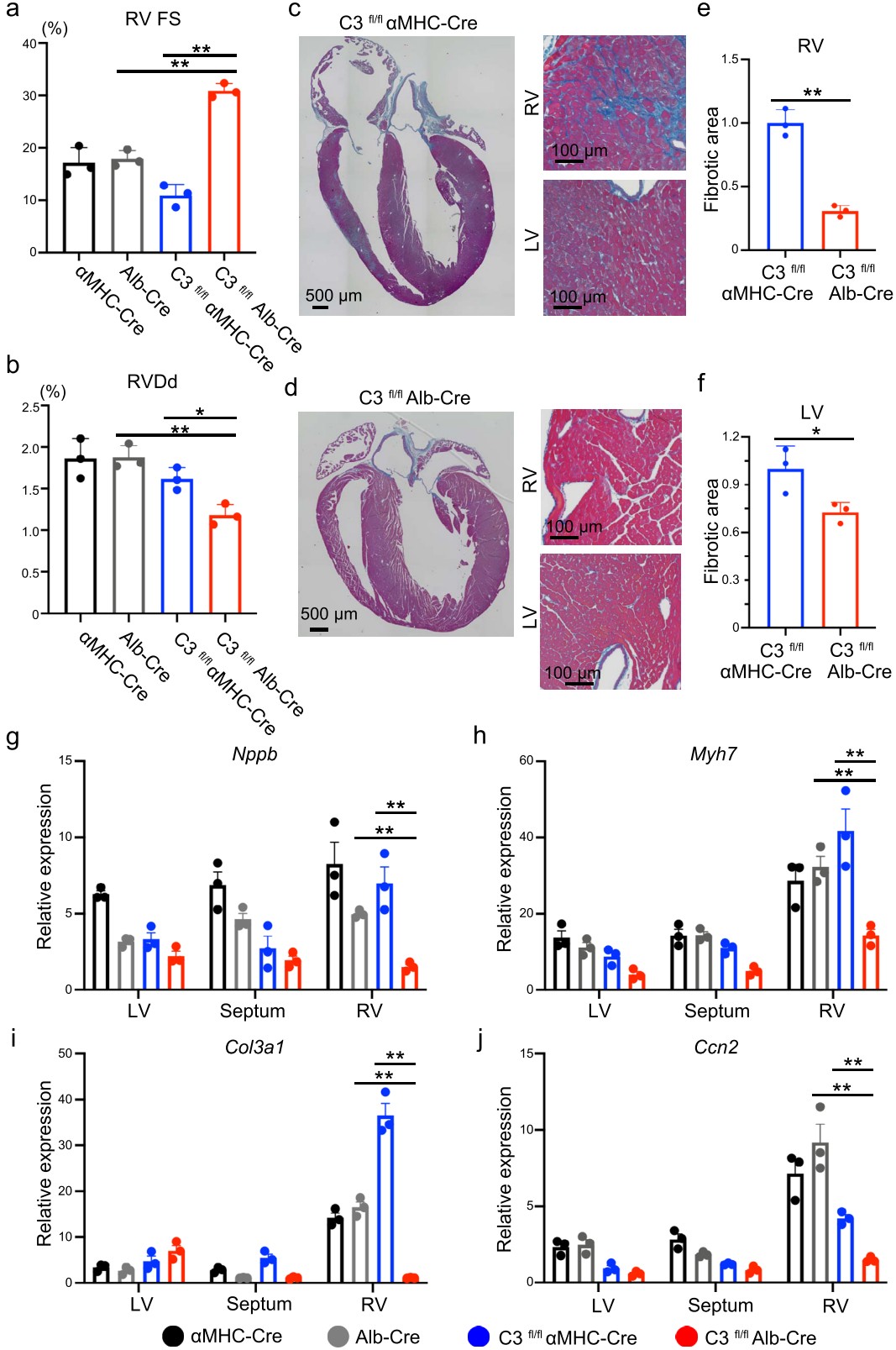

at present. It would be interesting to develop C3-, C3a-, or CFD-targeted drugs for RV failure treatment.

## Methods

### Animal

This study conformed to the Guide for the Care and Use of Laboratory Animals published by the US National Institutes of Health (Publication no. 85-23, revised 1996), and the study protocol was approved by the Institutional Animal Care and Use Committee at the Keio University School of Medicine. To generate *C3*-flox mice, a targeting vector was constructed as follows: a 1.1 kb DNA fragment carrying exons 24 and 25 of the *C3* gene was amplified by polymerase chain reaction (PCR) and inserted between the *Kpn*I sites of the middle entry clone (pDME-1). This clone consisted of a DNA fragment of pgk promoter-driven

**Fig. 2 | The liver-specific C3 deletion rescues pulmonary artery constriction (PAC)-induced right ventricular dysfunction in mice. a, b** Measured values obtained from the echocardiogram in α-myosin heavy chain promoter-driven Cre (αMHC-Cre) PAC, and albumin promoter-driven Cre (Alb-Cre) PAC, C3 floxed αMHC-Cre (C3$^{fl/fl}$ αMHC-Cre) PAC, and C3 floxed Alb-Cre (C3$^{fl/fl}$ Alb-Cre) PAC mice ($n = 3$, **a** $p = 0.0002$, $p < 0.0001$, **b** $p = 0.0015$, $p = 0.0147$). The right ventricle (RV) contractile function (right ventricular fractional shortening [RV FS]) and RV size (right ventricular end-diastolic diameter [RVDd]) were evaluated. Data are presented as mean ± standard deviation (SD). **c** Representative images of Azan staining of the heart in C3$^{fl/fl}$ αMHC-Cre PAC mice ($n = 3$). **d** Representative images of Azan staining of the heart in C3$^{fl/fl}$ Alb-Cre PAC mice. LV, left ventricle ($n = 3$).

**e, f** Quantified fibrotic area of the RV and LV in C3$^{fl/fl}$ αMHC-Cre and C3$^{fl/fl}$ Alb-Cre PAC model mice ($n = 3$, $p = 0.0005$, $p = 0.039$). Data are presented as mean ± SD. **g–j** qRT-PCR analysis of the expression of heart failure markers (*Nppb* and *Myh7*) and fibrotic markers (*Col3a1* and *Ccn2*) in the LV, ventricular septum, and RV of αMHC-Cre PAC, and Alb-Cre PAC, C3$^{fl/fl}$ αMHC-Cre PAC, and C3$^{fl/fl}$ Alb-Cre PAC mice ($n = 3$, **g** $p < 0.0001$, $p = 0.0003$, **h** $p < 0.0001$, $p < 0.0001$, **i** $p < 0.0001$, $p < 0.0001$, **j** $p < 0.0001$, $p = 0.0012$). Data are presented as mean ± standard error of the mean. In qRT-PCR analysis, expression of target genes was normalised to that of *Gapdh*. Significance was assessed using a two-tailed unpaired Student's *t*-test. *$p < 0.05$; **$p < 0.01$.

*Neo*-poly(A) flanked by two *frt* sites, and two *loxP* sequences located 123 bp upstream of exon 24 and 108 bp downstream of exon 25, respectively. The 5.8 kb upstream and 7.2 kb downstream homologous genomic DNA fragments were subcloned into the 5′ entry clone (pD5UE-2) and 3′ entry clone (pD3DE-2), respectively. For targeting vector assembly, the three entry clones were recombined into a destination vector plasmid (pDEST-DT; containing a cytomegalovirus enhancer/chicken actin (CAG) promoter-driven diphtheria toxin gene) using MultiSite Gateway Technology (Thermo Fisher Scientific). Homologous recombinant embryonic stem (ES) clones were identified by Southern blot and PCR analyses. ES cell culture and chimeric mice generation were performed as previously described[56].

C3$^{-/-}$ mice (C57BL/6 N background), α-myosin heavy chain promoter-driven Cre mice (αMHC-Cre) and albumin promoter-driven Cre mice (Albumin-Cre; C57BL/6 background) were purchased from the Jackson Laboratory. αMHC-Cre mice and Albumin-Cre mice were backcrossed to the C57BL/6 N background. Cardiac-specific C3-knockout mice were generated by crossing C3$^{flox/flox}$ mice with αMHC-Cre mice. Liver-specific C3$^{-/-}$ mice were generated by crossing C3$^{flox/flox}$ mice with Albumin-Cre mice. Cfd$^{-/-}$ mice (C57BL/6 N background) were generated using the CRISPR/Cas9 system, as described previously[23]. In all genetically modified mice, genotypes were confirmed by PCR analysis (please see Supplementary Table 1 for list of primer sequences).

### Establishment of the pulmonary artery constriction (PAC) mouse model
PAC surgery was performed on 8-week-old C57BL6 mice as previously described[15]. The surgery was performed using a ventilator to acquire passive respiration, and anaesthesia was induced with 2–4% isoflurane. The main trunk of the pulmonary artery was constricted using a 25-gauge blunt needle as the calibrator. The pulmonary artery blood flow was detected by Doppler echocardiogram analysis 2 weeks after surgery. The sham operation followed the same procedure, except that the pulmonary artery was not constricted. Mice were examined for the subsequent analyses 2–4 weeks after surgery.

### Establishment of the transverse aortic constriction (TAC) mouse model
TAC surgery was performed on 8-week-old mice, as previously described[15]. The surgery was performed using a ventilator to acquire passive respiration, and anaesthesia was induced with 2–4% isoflurane. The aorta was constricted using a 27-gauge blunt needle as the calibrator. The innominate artery and left common carotid artery blood flows were detected by Doppler echocardiogram analysis 1 week after surgery. The sham operation followed the same procedure, except that the aorta was not constricted. Mice were examined for the subsequent analyses 4 weeks after surgery.

### Echocardiography
Mice were anaesthetised using isoflurane, and echocardiography was performed using ultrasonography (Vevo 2100 system, Visual Sonics, Toronto, Canada) with a 30 MHz probe. Stable images were obtained in the M-mode, B-mode, and Doppler Mode. The LV inner dimension

and fractional shortening were measured using the M-mode. The RV inner dimension and fractional shortening were measured using the B-mode. The pulmonary arterial pressure gradient was obtained using the Doppler Mode. Heart rate did not differ significantly among the experimental groups during the echocardiographic assessments. All analyses were performed in a blinded manner with respect to the mice genotype.

### In vivo electrophysiological study (EPS)
An in vivo EPS was performed 2–4 weeks after PAC or sham operation. Mice were anaesthetised with a mixture of 0.3 mg/kg medetomidine, 4.0 mg/kg midazolam, and 5.0 mg/kg butorphanol. The heart rate was maintained at 400–500 bpm during the EPS. Programmed electrical stimulations (burst pacing and extrastimulus pacing) were delivered from the right ventricle using 1.1-Fr electrophysiology catheters (EPR-800, Millar, Houston, TX, USA), and electrocardiograms were recorded by electrodes placed on the extremities.

### In vivo hemodynamic study
LV and RV function were assessed by pressure catheter in vivo[57]. Mice were anesthetized using isoflurane. The right carotid vein was exposed, and a 1.4-Fr pressure catheter (SPR-839; Millar Instruments, Houston, TX) was inserted into the RV while recording in digital form (MPVS PL3508 PowerLab 8/35, ADInstruments) at the acquisition rate of 2 kHz for analysis (LabChart 8 pro, ADInstruments). Next, the left carotid artery was exposed, and a 1.4-Fr pressure catheter was inserted into the LV while recording LV pressure. All values were averaged over five consecutive cardiac cycles during stable phase of the respiratory cycle.

### Administration of the C3a receptor (C3aR) antagonist SB290157
C3aR antagonist SB290157 (Sigma-Aldrich) was administered (10 mg/kg body weight/day) using an Alzet micro-osmotic pump (model 2002). Alzet micro-osmotic pumps were implanted subcutaneously into the intrascapular region of mice on the day following PAC operation. During the pump implantation, the mice were anaesthetised with isoflurane.

### Transcriptome analysis
Mice were deeply anaesthetised and sacrificed by decapitation. Sternotomy was performed, cold phosphate-buffered saline (PBS) was perfused into the right and left ventricles, and the hearts were extirpated and separated into the right ventricle, left ventricle, and ventricular septum. The separated cardiac tissues were immediately frozen in liquid nitrogen. Frozen heart tissues were broken into powder using Cryopress (CP-100W, Microtec–Nichion Co. Ltd, Funabashi, Japan). These powders were dissolved in TRIzol reagent (Invitrogen), and total RNA was extracted. Similarly, cultured neonatal rat ventricular cardiomyocytes (NRVCs) were washed with cold PBS and immediately dissolved in TRIzol reagent. Transcriptome analysis was performed using SurePrint G3 Mouse Microarray 8X60 K ver.2.0 (Agilent Technologies Inc., Santa Clara, CA, USA) and SurePrint G3 Rat GE 8 × 60K ver. 2.0 (Agilent Technologies Inc.) The chips were scanned using Agilent Scanner G2505C (Agilent Technologies Inc.). The data

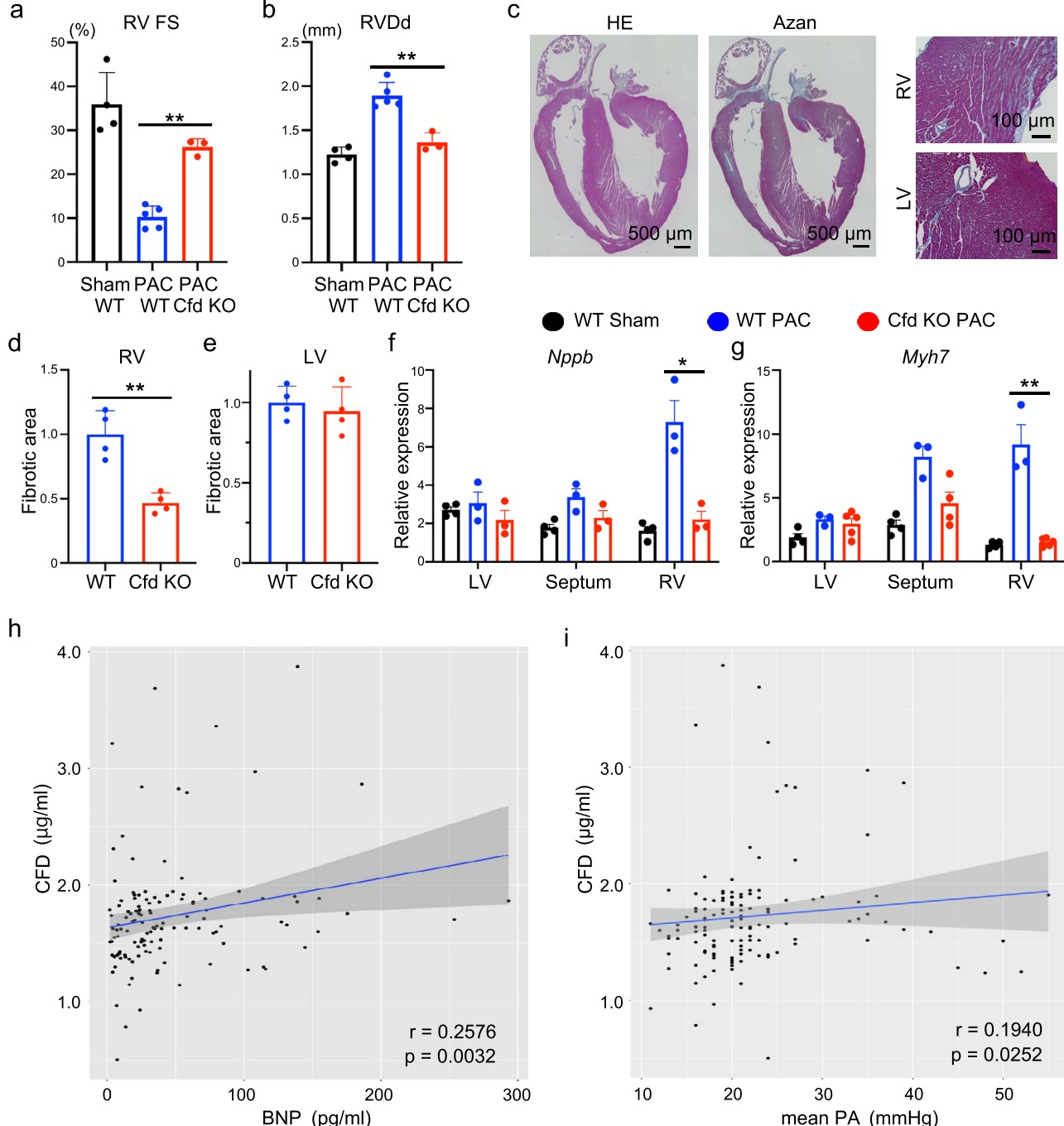

**Fig. 3 | Complement factor D (Cfd) plays an important role in mice with pulmonary artery constriction (PAC)-induced right ventricular (RV) dysfunction and patients with RV failure. a, b** Measured values obtained from the echocardiogram in wild type (WT) sham, WT PAC mice, and *Cfd* knockout (Cfd KO) PAC mice ($n = 4, 5, 3, p < 0.0001, p = 0.0017$). The right ventricle (RV) contractile function (right ventricular fractional shortening [RV FS]) and RV size (right ventricular end-diastolic diameter [RVDd]) were evaluated. Data are presented as mean ± standard deviation (SD). **c** Representative images of hematoxylin-eosin (HE) staining and Azan staining of the heart in Cfd KO PAC mice. LV, left ventricle ($n = 4$). **d, e** Quantified fibrotic area of the RV and LV in WT PAC and Cfd KO PAC model mice ($n = 4$). Data are presented as mean ± SD. **f, g** qRT-PCR analysis of the expression of heart failure markers (*Nppb* and *Myh7*) in the LV, ventricular septum, and RV of WT sham, WT PAC, and Cfd KO PAC mice ($n = 3$–5, $p = 0.0135$,

$p = 0.0005$). Data are presented as mean ± standard error of the mean (SEM). In qRT-PCR analysis, expression of target genes was normalised to that of *Gapdh*. Significance was assessed using a two-tailed unpaired Student's *t*-test. *$p < 0.05$; **$p < 0.01$. **h** Scatter plots showing the correlation between the CFD concentration and B-type natriuretic peptide (BNP) concentration in the overall cohort ($n = 129$; mean age = $66.5 ± 15.1$ years; 69.8% women). Spearman correlation coefficient and two-tailed p-value are shown. Linear regression line (blue line) with 95% confidence intervals (grey area) is represented. **i** Scatter plots showing the correlation between the CFD concentration and mean pulmonary artery (PA) pressure in the overall cohort ($n = 133$; mean age = $66.1 ± 15.3$ years; 70.7% women). Spearman correlation coefficient and two-tailed p-value are shown. Linear regression line with 95% confidence intervals is represented.

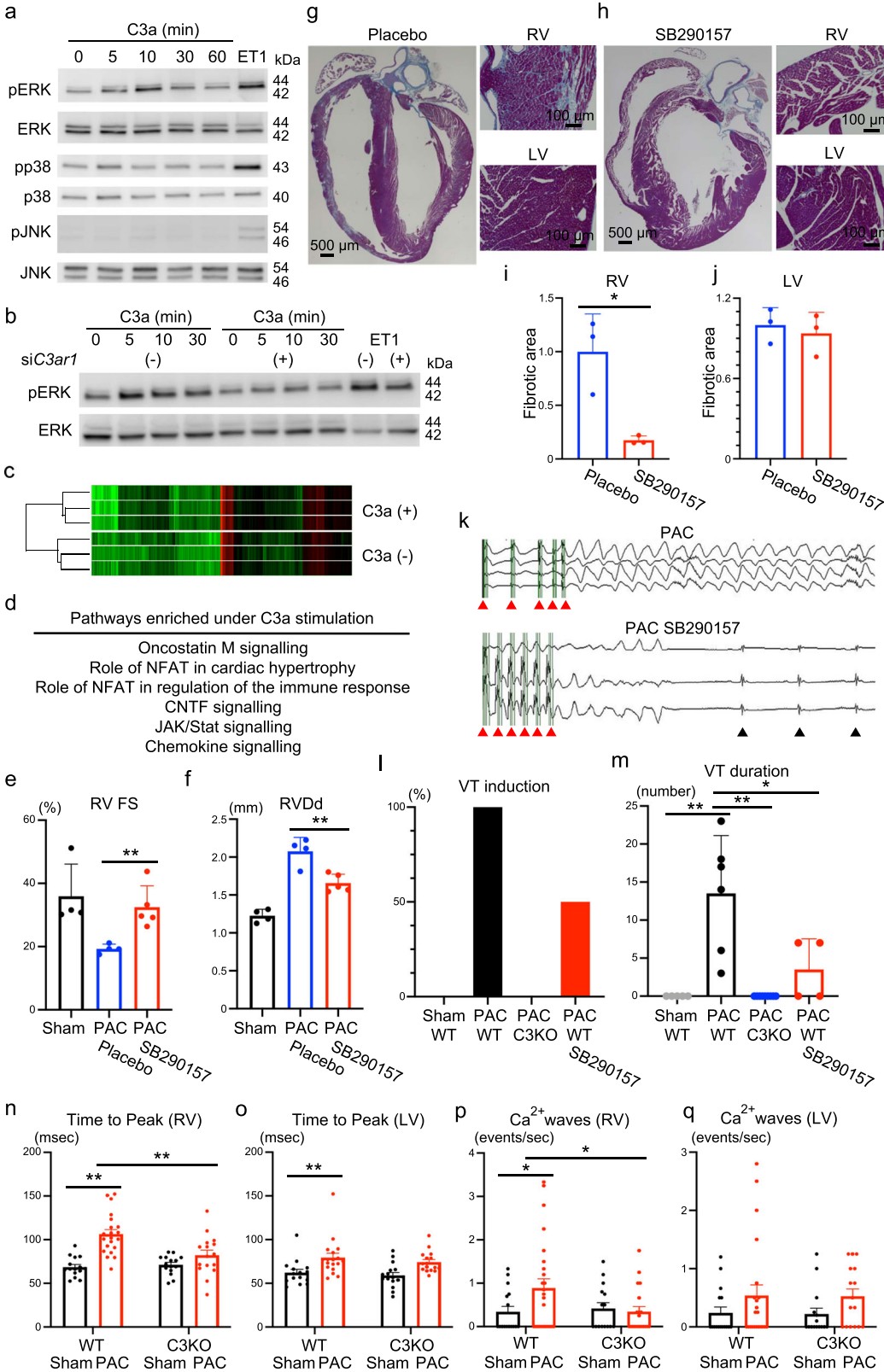

were processed and analysed using the GeneSpring software (v.14.1.1, Agilent Technologies Inc.). Statistical significance was assessed using analysis of variance (ANOVA) ($p < 0.05$) and corrected for multiple-group comparisons using Tukey's HSD. Significant differentially expressed genes were captured, and the expression patterns of these genes were analysed for pathway analysis using the Ingenuity Pathway Analysis software (Ingenuity Systems, www.ingenuity.com).

**Histology**

Mice were euthanized by cervical dislocation. The heart was immediately perfused with PBS and fixed with 10% neutral buffered formalin. The tissues were then embedded in paraffin. Sections were cut to a thickness of 4 μm and stained with hematoxylin and eosin. The connective tissue was visualised by Azan staining. For immunostaining, the hearts were perfused with PBS and fixed with 4% paraformaldehyde.

**Fig. 4 | C3a receptor blockade attenuates right ventricular (RV) failure and ventricular arrhythmia. a** Western blotting of phospho-ERK (pERK), total ERK (ERK), phospho-p38 (pp38), total p38, phospho-JNK (p-JNK), and total JNK ($n = 3$) in neonatal rat ventricular cardiomyocytes (NRVCs) treated with C3a in a time-dependent manner. Endothelin 1 (ET1) was used as a positive control. **b** Western blotting of pERK and ERK in the presence of *C3ar1* siRNA ($n = 3$). **c** Global gene expression heatmap of NRVCs in the presence of C3a ($n = 3$). **d** Pathway enrichment analysis using genes differentially expressed under recombinant C3a protein stimulation. **e, f** Measured values of the echocardiogram in wild type (WT) sham, WT pulmonary artery constriction (PAC), and WT PAC mice treated with SB290157 ($n = 4–5$, $p = 0.007$, $p = 0.004$). The Right ventricular (RV) fractional shortening [RV FS] and RV end-diastolic diameter [RVDd] were evaluated. Data are presented as mean ± standard deviation (SD). **g, h** Representative Azan staining images of WT PAC mice under placebo or SB290157 treatment ($n = 3$). LV, left ventricle.

**i, j** Quantified fibrotic area of the RV and LV in WT PAC mice under placebo or SB290157 treatment ($n = 3$, $p = 0.0156$, $p = 0.6217$). Data are presented as mean ± SD. **k** Representative electrocardiogram showing ventricular tachycardia (VT) after electrical program stimulation (red arrowhead) in WT PAC mice under SB290157 treatment. Black arrowhead shows normal ventricular beats. **l, m** The percentages of VT induction and VT duration after electrical program stimulation in WT sham, WT PAC, *C3* knockout (C3KO) PAC, and WT PAC mice treated with SB290157 ($n = 4–9$, $p < 0.001$, $p < 0.001$, $p = 0.044$). Data are presented as mean ± SD. **n–q** Measured values of $Ca^{2+}$ transients in singled cardiomyocytes from RV and LV of sham and PAC models of WT and C3KO mice. Time to peak and $Ca^{2+}$ wave frequency ($n = 15–26$, **n** $p < 0.001$, $p = 0.0017$, **o** $p = 0.0054$, **p** $p = 0.020$, $p = 0.0197$) were evaluated. Data are presented as mean ± standard error of the mean. Significance was assessed using a two-tailed unpaired Student's *t*-test. *$p < 0.05$; **$p < 0.01$.

Tissues were placed in 10 and 20% sucrose in PBS until they sank (1–2 h) and then in 30% sucrose in PBS overnight. For the detection of C3d, the samples were incubated with mouse monoclonal anti-C3d (3d29, Creative BioLabs) primary antibody for 1 h at 37 degrees. Antibody binding was detected using the horseradish peroxidase-conjugated anti-mouse IgG antibody (#NA931, 1:200, GE Healthcare Life Sciences), followed by incubation with diaminobenzidine (7411-49-6, Wako Pure Chemical Industries Ltd., Osaka, Japan). Sections were then counterstained with Mayer's hematoxylin (Burlingame, CA, USA). The fibrotic area was determined using the ImageJ software as previously described[58].

### Cell culture

Primary cultures of neonatal rat ventricular cardiomyocytes (NRVCs) were prepared as described previously[59]. After 1 h of serum starvation (1% fetal bovine serum), NRVCs were stimulated with C3a recombinant protein (8085-C3-025, R&D SYSTEMS). siRNAs were transfected into cells using Lipofectamine3000 (Invitrogen) according to the manufacturer's protocol. The siRNA for *C3ar1* (s136363) was purchased from Thermo Fisher Scientific. Control siRNA and scrambled siRNA (4390846) were purchased from Life Technologies.

### Isolation of adult cardiomyocytes

The process of adult cardiomyocyte isolation in mice was basically performed as described previously[60]. Briefly, the heart is quickly removed from a well-sedated mouse and cannulated in a Langendorff system. And the hearts were perfused into the coronary artery via the aorta with Cell Isolation Buffer containing Collagenase Type 4 (#CLS4, Worthington Biochemical Corporation) (1 mg/mL). And the left and right ventricles were separated and minced finely to isolate the cardiomyocytes. The pellets were then centrifuged at 300 rpm for 5 min, resuspended in Tyrode's solution, and stored at 37 degrees.

### Real-time quantitative reverse transcription PCR (qRT-PCR) analysis

In brief, total RNA was purified using the RNeasy Mini Kit (Qiagen). RNA samples were treated with gDNA Remover (Toyobo) to remove genomic DNA contamination. One microgram of DNase-treated RNA was used for first-strand complementary DNA (cDNA) synthesis using the ReverTra Ace qPCR RT Kit (Toyobo) and oligo dT20 primers. qPCR was performed using Fast SYBR Green Master Mix (Thermo Fisher Scientific). The primer sequences are listed in Supplementary Table 2.

### Western blot analysis

Briefly, the NRVCs were lysed using the ULTRARIPA kit A solution (BioDynamics Laboratory Inc., Tokyo, Japan) containing 1% protease inhibitor (P8340, Sigma-Aldrich) and 1% phosphatase inhibitor (160-24371, Wako Pure Chemical Industries Ltd.). The cell lysates were centrifuged at $12,000 \times g$, and the protein concentration in the supernatant was determined using the BCA Protein Assay Kit (TaKaRa Bio Inc., Japan). Total proteins were resolved in 15% sodium dodecyl

sulfate polyacrylamide gels under reducing conditions and electrophoretically transferred onto polyvinylidene difluoride membranes using the iBlot Dry Blotting system (Thermo Fisher Scientific). The membranes were incubated with Blocking One (Nacalai Tesque, Kyoto, Japan) for 30 min to block nonspecific binding sites. The polyvinylidene difluoride membranes were then incubated overnight with the primary antibodies diluted in Hikari Signal Enhancer Solution (Nacalai Tesque) at 4 degrees. Next, the membranes were incubated with horseradish peroxidase-conjugated anti-rabbit IgG (NA934, GE Healthcare) secondary antibody for 1 h at room temperature. Labelled proteins were visualised using an enhanced chemiluminescence kit (Nacalai Tesque) according to the manufacturer's instructions. Primary antibodies related to MAP kinesis were rabbit anti-p44/42 MAPK (4370, Cell Signalling Technology), anti-phospho-p44/42 MAPK (4695, Cell Signalling Technology), rabbit anti-p38 (9212, Cell Signalling Technology), rabbit anti-phospho p38 (9211, Cell Signalling Technology), rabbit anti-SAPK/JNK (9252, Cell Signalling Technology), and rabbit anti-phospho-SAPK/JNK (9251, Cell Signalling Technology).

### Measurement of $Ca^{2+}$ Imaging

Briefly, isolated cardiomyocytes were loaded with Tyrode solution containing 1 µM Fluo-4AM (F14217, Themo Fisher Scientific) and stained for 20 min. The loaded cardiomyocytes were then resuspended in normal Tyrode solution containing 1 mM CaCl2 and kept at 37 degrees. Electric field stimulation (DPS-007, DIA Medical System Co) was applied to loaded cardiomyocytes. During this process, $Ca^{2+}$ transients, $Ca^{2+}$ waves, and $Ca^{2+}$ sparks measurements were captured on movies using an All-In-One microscope (BZ-9000, Keyence). The movies were analyzed using Image J. The specific $Ca^{2+}$ imaging measurement protocols are as follows. $Ca^{2+}$ transient was measured as the change in fluorescence intensity of fluo4 during 5 Hz electric field stimulation for approximately 5 s. $Ca^{2+}$ waves and $Ca^{2+}$ sparks were analyzed as the change in fluorescence intensity emitted from isolated cardiomyocytes during 15 s after field stimulation. $Ca^{2+}$ waves were counted as fluorescence intensity in 90% of the surface area of cardiomyocytes. The $Ca^{2+}$ sparks were counted as a localized $Ca^{2+}$ as fluorescence intensity in the cardiomyocytes along a 100 µm line. The parameters of Fmax/F0, Time to Peak, and RT50 were calculated from the $Ca^{2+}$ transients. Fmax/F0 was calculated by defining F0 as the fluorescence intensity of Fluo4 at rest before field stimulation and Fmax as the maximum fluorescence intensity of Fluo4 during field stimulation. Time to Peak was calculated as the difference between the time of the first electric field stimulation and the time of the first Fmax. RT50 was calculated as the difference between the time at Fmax and the time at which the fluorescence intensity decreased to 50% of Fmax.

### Human sample correction

We retrospectively evaluated patients with chronic thromboembolic pulmonary hypertension (CTEPH) and pulmonary arterial hypertension (PAH) from April 2016 to November 2020 according to our

Article

inclusion criteria: (1) age ≥18 years, (2) subjected to right heart catheterisation (RHC), and (3) availability of plasma samples. Thus, 133 patients, including 128 with CTEPH and 5 with PAH, were enrolled for the main analysis (mean age = 66.1 ± 15.3 years; 70.7% women). All patients underwent RHC using a 6 or 7 Fr Swan-Ganz catheter (Swan-Ganz CCO CEDV, Edwards Lifesciences, Irvine, CA, USA). The mean pulmonary artery (PA) pressure was measured using RHC. The present study was approved by the Ethics Committee of Keio University Hospital (approval no. 20140203). Written informed consent was obtained from all participants in this human study and no compensation was given to participants. Total RNAs from the left and right ventricle of adult humans were purchased from Biochain (cat. no. R1234138-50, and R1234139-50). RT-PCR analysis for human samples was carried out in technical replicates (Fig. 1d–f).

### Enzyme linked immunosorbent assay (ELISA)

Plasma C3 and CFD levels were measured using the human C3 (HK366) and human complement factor D (HK343) ELISA kits (Hycult Biotech, Uden, Netherlands). Samples, reagents, and buffers were prepared according to the manufacturer's protocol). All laboratorial analyses were performed in a blinded manner.

### Statistical analyses

Values are presented as the mean ± standard error of the mean or as mean ± standard deviation. The significance of the differences between two means was evaluated using unpaired and paired $t$-tests. Spearman's correlation test was performed to compare the plasma concentrations of C3 and CFD with the RHC data. Statistical significance was set at $P < 0.05$.

### Reporting summary

Further information on research design is available in the Nature Research Reporting Summary linked to this article.

## Data availability

The microarray data generated in this study have been deposited in the GEO database under accession code GSE183503 and GSE183504. All the other data supporting the findings of this study are available within the article and its Supplementary Information files. Source data are provided with this paper.

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

## Acknowledgements

We thank all the members of our laboratory for their assistance. This research was supported by Grants-in-Aid for Scientific Research (JSPS KAKENHI, grant numbers 16H05304 (SY), 16K15415 (SY), 18K08047 (HY), 19H03622 (SY), 20H03678 (SY), 20K08461 (SY), 20K08193 (SY)), the SENSHIN Medical Research Foundation (SY), the Fukuda Foundation for Medical Technology (HY), the Daiwa Securities Health Foundation (HY), the Miyata Cardiac Research Promotion Foundation (HY), and Kawano Masanori Memorial Public Interest Incorporated Foundation for Promotion of Pediatrics (HY).

## Author contributions

S.I. and S.Y. designed the experiments. S.I., H.H., H.Y., D.K., Y.A., T.N., M.M., J.K., T.K., M.K., Y.K., S.K., A.K., M.L., M.S., G.Y., C.M., T.S., T.Y., Y.S., T.H., M.K., Y.K., K.Suzuki., K.I., M.A., M.O., H.T., M.Y., K.Sakimura., M.Y., and S.Y performed the experiments. M.M., Y.S., T.H., M.K., and T.K collected the human samples. K.I. and H.Y. conducted the mouse electrophysiological study. H.Y. conducted Ca$^{2+}$ imaging study. H.T. and M.Y. generated the *Cfd* knockout mice. K.Suzuki., M.A., K.Sakimura. and M.Y. generated the conditional *C3* knockout mice. Y.F. and K.F. supervised the study. S.I. and S.Y. wrote the manuscript. All authors read and approved the final manuscript.

## Competing interests

K.F. is a founding scientist funded by the SAB of Heartseed Co., Ltd. All the other authors declare no competing interests.

## Additional information

---

[1]Department of Cardiology, Keio University School of Medicine, 35 Shinanomachi, Shinjuku-ku, Tokyo 160-8582, Japan. [2]Division of Cardio-Vascular Medicine, Department of Internal Medicine, Kurume University School of Medicine, 67 Asahi-machi, Kurume, Fukuoka 830-0011, Japan. [3]Center for Preventive Medicine, Keio University School of Medicine, 35 Shinanomachi, Shinjuku-ku, Tokyo 160-8582, Japan. [4]Department of Healthcare Information Management, The University of Tokyo Hospital, 7-3-1 Hongo, Bunkyo-ku, Tokyo 113-8655, Japan. [5]Second Department of Internal Medicine, University of Occupational and Environmental Health, 1-1 Iseigaoka, Yahatanishi-ku, Kitakyushu, Fukuoka 807-8555, Japan. [6]Department of Physiology, Keio University School of Medicine, 35 Shinanomachi, Shinjuku-ku, Tokyo 160-8582, Japan. [7]Department of Cardiology, National Defense Medical College, 3-2 Namiki, Tokorozawa, Saitama 359-8513, Japan. [8]Department of Cardiology, International University of Health and Welfare, Mita Hospital, 1-4-3 Mita Minatoku, Tokyo 108-8329, Japan. [9]Department of Animal Model Development, Brain Research Institute, Niigata University, 1-757 Asahimachi-Dori, Chuo-ku, Niigata 951-8585, Japan. [10]Department of Life Sciences and Bioethics, Graduate School of Medical and Dental Sciences, Tokyo Medical and Dental University, 1-5-45 Yushima, Bunkyo-ku, Tokyo 113-8510, Japan. ✉e-mail: yuasa@keio.jp

