## [Peer Review File · Nature Communications]

REVIEWER COMMENTS

Reviewer #1 (Remarks to the Author):

This is a very interesting study detailing the role of the complement pathway in the setting of right ventricular failure. Even though left ventricular failure has been very well studied and the molecular mechanisms of action have been well established, cellular and molecular mechanisms of right ventricular failure are not well understood. The authors show that the right ventricle predominantly expresses alternative complement pathway-related genes, including *Cfd* and *C3aR1*. They generated a systemic complement 3 (C3)-knockout attenuated RV dysfunction and fibrosis in the mouse RV failure model induced by pulmonary arterial banding. C3 conditional knockout mice revealed that the liver-derived, but not the heart-derived, C3 played a crucial role in RV failure. Complement factor D (*Cfd*) knockout mice also showed attenuation of RV failure in the setting of PAC. C3a directly regulated expression of several genes through the C3a receptor (C3aR) in cultured cardiomyocytes. The C3aR antagonist dramatically improved RV dysfunction in the RV failure model mice. The data generated by the authors are of high quality and convincingly show the crucial role of the C3-*Cfd*-C3aR axis in RV failure and the potential therapeutic targets for RV failure.

I have a few comments for the authors.

1. In the PAC model which the authors use effectively there are no measurements of what happens to trans-pulmonary valve pressures in the different KO models they use. That would be useful to assess the cellular and molecular changes
2. The authors measure LV fractional shortening but there is little information on more direct RV functional assessments in vivo
3. It would be useful to attempt isolating single cardiac myocytes from the RV and LV and measure cellular functions to understand the differential effects of liver versus cardiac C3 KO
4. It would be important to validate the complement changes in human RV/LV samples if the authors have access to that type of tissue.
5. The Pulmonary Vascular Disease Phenomics (PVDOMICS) program has reported the importance of the complement pathway in the disease state and it may be worthwhile to reference.

Reviewer #2 (Remarks to the Author):

The manuscript describes studies on right ventricular (RV) failure, which plays a critical role in heart failure. Whereas the mechanisms underlying left ventricular (LV) failure have been proposed, and drugs for LV failure have been developed, not much is known about the mechanism of RV failure, and there is no specific therapy.

The authors initiate the studies by studying the difference in expression of genes in various parts of the heart. They discover a number of differences. They choose to concentrate on the finding that the right ventricle expressed a higher level of alternative complement pathway-related genes, including *Cfd* and *C3aR1*. With this in mind, the authors test whether complement factor C3 influences RV dysfunction. They find that mice lacking C3 have attenuated RV dysfunction and fibrosis in a mouse RV failure model. If they instead use C3 conditional knockout mice, they find that the liver-derived C3 (being absent in C3 floxed albumin promoter-driven Cre (C3^{fl/fl} Alb-Cre) mice) but not heart-derived C3 (being absent in C3 floxed α -myosin heavy chain promoter-driven Cre (C3^{fl/fl} α MHC-Cre) mice) played a role in RV failure. They further study mice lacking

the enzyme Factor D (Cfd) of the complement system and report that these mice showed attenuation of RV failure.

They test the level of complement factor D in a cohort of patients in patients with chronic RV failure and claim that the plasma concentration of CFD is correlated with the severity of RV failure. This is not the case if testing for complement factor C3.

The authors test if C3a (a fragment of C3 produced when the enzyme Factor Bb cleaves C3 into C3b and C3a) regulates the expression of genes through the C3a receptor (C3aR) in cultured cardiomyocytes. A C3aR antagonist improved RV dysfunction in a RV failure model mice.

They conclude that they have demonstrated a role of the complement system via a C3-Cfd-C3aR axis in RV failure.

Although relatively few animals are included in each group when performing the experiments, I am impressed with the animal studies.

Specific comments.

The authors claim that the results revealed by the studies of mice are translatable to the human situation. The authors report that scatter plots (Fig. 3h and i) show a significant correlation between the CFD concentration and B-type natriuretic peptide (BNP) concentration in the overall cohort ($r = 0.26$, $n = 128$) and a significant correlation between the CFD concentration and mean pulmonary artery (PA) pressure in the cohort ($r = 0.19$, $n = 128$). As is clear from the numbers, these are very weak correlations. I suggest the authors instead report this as the coefficient of correlation (r) with the 95% confidence interval given. If the authors believe in a linear link, they should use Person's correlation. If they just wish to examine if two parameters follow each other, it is better to use Spearman's rho.

The authors write: "These data suggest that CFD was involved in the pathogenesis of human RV failure.". It is not correct to claim causality on the basis of a scatter plot. The data for the CFD concentrations could not be found in the files with source data.

Does CRE- influence the outcome of the results? A control of MHC-CRE mice crossed with wildtype animals should have been included in the studies, e.g. in the graphs in Fig. 2. It may be an important control as others have reported an influence of MHC-CRE on heart functions in mice, e.g., as described in Cardiac-Specific Cre Induces Age-Dependent Dilated Cardiomyopathy (DCM) in Mice Taha Rehmani, Maysoon Salih, and Balwant S. Tuana *Molecules*. 2019 Mar; 24(6): 1189. doi: 10.3390/molecules24061189. At least there should be a discussion on the ongoing debate on the occurrence of cardiac side effects caused by unspecific Cre activity or related to tamoxifen/oil overload.

The authors test for the presence of C3d by immunohistochemical staining the RV tissue of PAC model and sham-operated mice (Fig. 1g). The antibody that is used for the IHC is a polyclonal goat anti-C3d antibody. This antibody reacts with epitopes that are shared by C3, C3b, iC3b, and C3d. It is thus not known which form of C3 that is found in the IHC.

The authors use a small molecule C3aR antagonist – referred to as SB290157 - as a research tool to explore C3aR function. It is important to note/discuss that this compound may have other effects. This is, e.g. described in: The "C3aR Antagonist" SB290157 is a Partial C5aR2 Agonist. Xaria X. Li, Vinod Kumar, Richard J. Clark, John D. Lee, and Trent M. Woodruff. *Front. Pharmacol.*, <https://doi.org/10.3389/fphar.2020.591398>.

In the discussion section, the authors suggest that a possible influence of the complement system on RV failure could be linked to a role in protection from infections. The authors should also include other functions of the complement system in such

discussions, i.e., that complement is a complex innate immune surveillance system, playing a role in host homeostasis, inflammation, and in the defense against pathogens.

Response to Reviewers.

We thank the reviewers for careful reviewing and constructive comments, which helped us to improve the manuscript. According to the reviewer's comments, we performed additional experiments and revised the manuscript.

Reviewer #1 (Remarks to the Author):

This is a very interesting study detailing the role of the complement pathway in the setting of right ventricular failure. Even though left ventricular failure has been very well studied and the molecular mechanisms of action have been well established, cellular and molecular mechanisms of right ventricular failure are not well understood. The authors show that the right ventricle predominantly expresses alternative complement pathway-related genes, including Cfd and C3aR1. They generated a systemic complement 3 (C3)-knockout attenuated RV dysfunction and fibrosis in the mouse RV failure model induced by pulmonary arterial banding. C3 conditional knockout mice revealed that the liver-derived, but not the heart-derived, C3 played a crucial role in RV failure. Complement factor D (Cfd) knockout mice also showed attenuation of RV failure in the setting of PAC. C3a directly regulated expression of several genes through the C3a receptor (C3aR) in cultured cardiomyocytes. The C3aR antagonist dramatically improved RV dysfunction in the RV failure model mice. The data generated by the authors are of high quality and convincingly show the crucial role of the C3-Cfd-C3aR axis in RV failure and the potential therapeutic targets for RV failure.

I have a few comments for the authors.

1. In the PAC model which the authors use effectively there are no measurements of what happens to trans-pulmonary valve pressures in the different KO models they use. That would be useful to assess the cellular and molecular changes

Thank you for the comments. Here, we used pulmonary artery constriction (PAC) model. In that model, the main trunk of the pulmonary artery is constricted, which is distal site of pulmonary valve. In our study, it is technically difficult to measure pressure gradient at trans-pulmonary valve. But we conducted the echo study at the pulmonary artery constriction site in the different KO models. We could measure the pressure gradient through PA constriction site (Supplementary Figure 1k, Supplementary Figure 3c, Supplementary Figure 4a, and Supplementary Figure 6a). The pressure gradients are similar in all KO models before and after PAC, which suggests that comparable pressure is loaded at pulmonary valve and right ventricle.

2. The authors measure LV fractional shortening but there is little information on more direct RV functional assessments in vivo.

Thank you for the comments. It is important to directly measure right ventricular function. In the echo study, the parameters of RV function were assessed, such as RV fractional shortening and RV diameter (Figure 1j, k, Figure 2a, b, Figure 3a, b, Figure 4e, f). Additionally, we conducted in vivo hemodynamic study by pressure catheter. In the study, RV end-diastolic pressure was measured to assess the RV function. RV end-diastolic pressure was increased after PAC and C3KO mice attenuated the increase of end-diastolic pressure (Figure 1m). LV end-diastolic pressure was not changed by PAC in WT and C3KO mice (Supplementary Figure 1n).

3. It would be useful to attempt isolating single cardiac myocytes from the RV and LV and measure cellular functions to understand the differential effects of liver ver cardiac C3 KO

Thank you for the comments. In order to measure single cellular function, single adult cardiomyocytes were isolated by using Langendorff system. Single cardiomyocytes from RV and LV were separately obtained after either PAC or Sham operation in WT and C3KO mice, and calcium imaging was conducted to measure cellular functions (Figure 4n-q, Supplementary Figure 6l-q). Ca²⁺ transient observation study showed significant slower time to peak in WT PAC RV

cardiomyocytes, which was attenuated in C3KO PAC RV cardiomyocytes (Figure 4n). The frequency of Ca²⁺ waves was significantly elevated in WT PAC RV cardiomyocytes, which was attenuated in C3KO PAC RV cardiomyocytes (Figure 4p). These data suggest that the cellular function of RV cardiomyocytes is damaged by PAC and its damage is ameliorated by C3KO.

4. It would be important to validate the complement changes in human RV/LV samples if the authors have access to that type of tissue.

Thank you for the comments. We agreed with your opinion. It is important to validate the complement-related gene expression in human samples. We obtained humans samples and confirmed that the complement-related genes, *CFD*, *C3* and *C3ARI* are highly expressed in human RV samples (Figure 1d-f) in agreement with mouse study.

5. The Pulmonary Vascular Disease Phenomics (PVDOMICS) program has reported the importance of the complement pathway in the disease state and it may be worthwhile to reference.

Thank you for the comments. As you pointed out, previous report showed the importance of the complement pathway in the pathogenesis of pulmonary hypertension, especially in lung. In order to enrich the discussion, we revised the discussion and added the references (page 11, line 18-21).

Reviewer #2 (Remarks to the Author):

The manuscript describes studies on right ventricular (RV) failure, which plays a critical role in heart failure. Whereas the mechanisms underlying left ventricular (LV) failure have been proposed, and drugs for LV failure have been developed, not much is known about the mechanism of RV failure, and there is no specific therapy.

The authors initiate the studies by studying the difference in expression of genes in various parts of the heart. They discover a number of differences. They choose to concentrate on the finding that the right ventricle expressed a higher level of alternative complement pathway-related genes, including Cfd and C3aR1. With this in mind, the authors test whether complement factor C3 influences RV dysfunction. They find that mice lacking C3 have attenuated RV dysfunction and fibrosis in a mouse RV failure model. If they instead use C3 conditional knockout mice, they find that the liver-derived C3 (being absent in C3 floxed albumin promoter-driven Cre (C3fl/fl Alb-Cre) mice) but not heart-derived C3 (being absent in C3 floxed α -myosin heavy chain promoter-driven Cre (C3fl/fl α MHC-Cre) mice) played a role in RV failure. They further study mice lacking the enzyme Factor D (Cfd) of the complement system and report that these mice showed attenuation of RV failure.

They test the level of complement factor D in a cohort of patients in patients with chronic RV failure and claim that the plasma concentration of CFD is correlated with the severity of RV failure. This is not the case if testing for complement factor C3.

The authors test if C3a (a fragment of C3 produced when the enzyme Factor Bb cleaves C3 into C3b and C3a) regulates the expression of genes through the C3a receptor (C3aR) in cultured cardiomyocytes. A C3aR antagonist improved RV dysfunction in a RV failure model mice.

They conclude that they have demonstrated a role of the complement system via a C3-Cfd-C3aR axis in RV failure.

Although relatively few animals are included in each group when performing the

experiments, I am impressed with the animal studies.

Specific comments.

The authors claim that the results revealed by the studies of mice are translatable to the human situation. The authors report that scatter plots (Fig. 3h and i) show a significant correlation between the CFD concentration and B-type natriuretic peptide (BNP) concentration in the overall cohort ($r = 0.26$, $n = 128$) and a significant correlation between the CFD concentration and mean pulmonary artery (PA) pressure in the cohort ($r = 0.19$, $n = 128$). As is clear from the numbers, these are very weak correlations. I suggest the authors instead report this as the coefficient of correlation (r) with the 95% confidence interval given. If the authors believe in a linear link, they should use Person's correlation. If they just wish to examine if two parameters follow each other, it is better to use Spearman rho.

Thank you for the comments. We agreed with your important opinion. In our cohort, although there are significant correlations between CFD and BNP, and between CFD and PA pressure, its correlations are weak. As you suggested we showed here the coefficient of correlation (r) with the 95% confidence interval given (Figure 3h, i, Supplementary Figure 4f, g).

The authors write: "These data suggest that CFD was involved in the pathogenesis of human RV failure.". It is not correct to claim causality on the basis of a scatter plot.

Thank you for the comments. We agreed with your opinion that our human data could not show the causality, but simply show the correlation. We revised the manuscript in page 8, line 10-11.

The data for the CFD concentrations could not be found in the files with source data.

Thank you for the comments. We added the CFD concentration data in supplement file.

Does CRE- influence the outcome of the results? A control of MHC-CRE mice crossed with wildtype animals should have been included in the studies, e.g. in the graphs in Fig. 2. It may be an important control as others have reported an influence of MHC-CRE on heart functions in mice, e.g., as described in Cardiac-Specific Cre Induces Age-Dependent Dilated Cardiomyopathy (DCM) in Mice Taha Rehmani, Maysoon Salih, and Balwant S. Tuana *Molecules*. 2019 Mar; 24(6): 1189. doi: 10.3390/molecules24061189. At least there should be a discussion on the ongoing debate on the occurrence of cardiac side effects caused by unspecific Cre activity or related to tamoxifen/oil overload.

Thank you for your comments. As you pointed out, *Cre* gene may influence the phenotypes of disease state in experimental animal models. Therefore, we examined aMHC-Cre mice and Alb-Cre mice as controls for the echo study and RT-PCR analysis (Figure 2a, b, g-j, Supplementary Figure 3c-g). We confirmed the RV failure was significantly attenuated in the liver-specific C3 knockout mice, compared with aMHC-Cre mice, Alb-Cre mice and the cardiac-specific C3 knockout mice.

The authors test for the presence of C3d by immunohistochemical staining the RV tissue of PAC model and sham-operated mice (Fig. 1g). The antibody that is used for the IHC is a polyclonal goat anti-C3d antibody. This antibody reacts with epitopes that are shared by C3, C3b, iC3b, and C3d. It is thus not known which form of C3 that is found in the IHC.

Thank you for your comments. As you pointed out, a polyclonal anti-C3d antibody may show cross-react against other forms of C3. Therefore, we used a monoclonal antibody for C3d, which targets tissue-bound C3d in vivo (Figure 1l). We confirmed that C3d was detected in the right ventricle only after PAC.

The authors use a small molecule C3aR antagonist – referred to as SB290157 -

as a research tool to explore C3aR function. It is important to note/discuss that this compound may have other effects. This is, e.g. described in: The “C3aR Antagonist” SB290157 is a Partial C5aR2 Agonist. Xaria X. Li, Vinod Kumar, Richard J. Clark, John D. Lee, and Trent M. Woodruff. Front. Pharmacol., <https://doi.org/10.3389/fphar.2020.591398>.

Thank you for your important comments. We added the discussion about SB290157 which may have other effects (page 12, line 18-21).

In the discussion section, the authors suggest that a possible influence of the complement system on RV failure could be linked to a role in protection from infections. The authors should also include other functions of the complement system in such discussions, i.e., that complement is a complex innate immune surveillance system, playing a role in host homeostasis, inflammation, and in the defense against pathogens.

Thank you for your important comments. We added the discussion about the functions of the complement systems (page 12, line 8-13).

REVIEWERS' COMMENTS

Reviewer #1 (Remarks to the Author):

In their revised manuscript the authors validated the complement changes in human RV samples and conducted isolated cardiomyocyte studies from the RV that validated their conclusions.

Reviewer #2 (Remarks to the Author):

The authors have answered the questions asked by the reviewers and have made appropriate changes to the manuscript.